# Transcriptome Meta-Analysis of Triple-Negative Breast Cancer Response to Neoadjuvant Chemotherapy

**DOI:** 10.3390/cancers15082194

**Published:** 2023-04-07

**Authors:** Wei Zhang, Emma Li, Lily Wang, Brian D. Lehmann, X. Steven Chen

**Affiliations:** 1Department of Public Health Sciences, University of Miami Miller School of Medicine, Miami, FL 33136, USA; 2California Academy of Mathematics and Science, 1000 E Victoria St, Carson, CA 90747, USA; 3Sylvester Comprehensive Cancer Center, University of Miami Miller School of Medicine, Miami, FL 33136, USA; 4Department of Medicine, Vanderbilt University Medical Center, Nashville, TN 37232, USA; 5Vanderbilt-Ingram Cancer Center, Vanderbilt University Medical Center, Nashville, TN 37232, USA

**Keywords:** TNBC, neoadjuvant chemotherapy, biomarker, meta-analysis

## Abstract

**Simple Summary:**

This study aimed to identify genes associated with neoadjuvant chemotherapy (NAC) response and disease-free survival (DFS) of triple-negative breast cancer (TNBC) patients. We conducted a large-scale meta-analysis of gene expression data from multiple TNBC cohorts and analyzed four gene expression quadrants from an integrated analysis of NAC response and clinical outcomes. The study also highlighted the importance of incorporating both short-term and long-term clinical outcomes in NAC response evaluation. The findings provide insights into the complex molecular mechanisms underlying TNBC and may guide the development of personalized treatment strategies for TNBC patients.

**Abstract:**

Triple-negative breast cancer (TNBC) is a heterogeneous disease with varying responses to neoadjuvant chemotherapy (NAC). The identification of biomarkers to predict NAC response and inform personalized treatment strategies is essential. In this study, we conducted large-scale gene expression meta-analyses to identify genes associated with NAC response and survival outcomes. The results showed that immune, cell cycle/mitotic, and RNA splicing-related pathways were significantly associated with favorable clinical outcomes. Furthermore, we integrated and divided the gene association results from NAC response and survival outcomes into four quadrants, which provided more insights into potential NAC response mechanisms and biomarker discovery.

## 1. Introduction

Triple-negative breast cancer (TNBC) refers to a heterogeneous collection of tumors that lack expression of the estrogen receptor (ER), progesterone receptor (PR), and HER2 amplification. TNBC is more prevalent among younger patients and African American women, representing around 15% to 20% of all breast cancers [1,2]. The estimated number of newly diagnosed TNBC cases is around 60,000 each year in the United States, which is comparable to, or even larger than, patient cohorts for many other cancer types, such as pancreatic, rectal, prostate, and ovarian cancers [3]. TNBC tumors are generally larger in size, are of higher grade, have lymph node involvement at diagnosis, and are biologically more aggressive [4]. Despite having higher rates of clinical response to pre-surgical (neo-adjuvant) chemotherapy, TNBC patients have a higher rate of distant recurrence and a poorer prognosis than women with other breast cancer subtypes [1,4]. Fewer than 30% of women with metastatic TNBC survive five years, and almost all die of their disease despite adjuvant chemotherapy [1].

The molecular landscape of TNBC has been extensively studied, and molecular subtypes of TNBC based on transcriptome data have been identified and utilized by the research community [5,6,7,8,9,10,11]. However, the targeted treatment strategies based on molecular subtypes requires additional clinical trials to prove their efficacy [12]. The emerging therapeutic options, including immune checkpoint inhibitors (ICIs) [13,14], poly-ADP-ribosyl polymerase (PARP) inhibitors [15,16], and antibody–drug conjugates (ADCs) [17,18], are available for metastatic TNBC only at the current stage, and the standard neoadjuvant treatment for non-metastatic TNBC is still conventional chemotherapy [12]. Treatment of TNBC often includes neoadjuvant chemotherapy with a combination of anthracyclines, alkylating agents, platinum salts, and taxanes, which has proven to be effective in the treatment of TNBC for a subset of patients. Patients who experience a pathologic complete response (pCR) and show no evidence of disease in the surgical specimen following neoadjuvant chemotherapy (NAC) have significant improvements in both disease-free and overall survival compared with patients with the residual invasive disease [19], while those patients with the residual disease have a much poorer prognosis and are 6 times more likely to have a recurrence and 12 times more likely to die [20,21]. 

Neoadjuvant chemotherapy is the standard for newly diagnosed TNBC. While ~30% of patients respond and have favorable outcomes, there are a large portion of patients who must endure the toxicities associated with chemotherapy. The ability to identify those patients unlikely to respond could give clinicians guidance to enroll patients on alternative compounds in clinical trials. Currently, there is no effective way to identify these TNBC patients. Gene signatures have been made to guide chemotherapy decisions for breast cancer, such as MammaPrint, but all TNBC samples fall into the poor prognosis category and are predicted to be responsive to NAC [22]. A series of gene signatures have also been developed to predict the pathologic complete response (pCR) of NAC in TNBC [23,24,25,26,27,28,29,30,31]. Although some meaningful patterns have emerged, such as the association of immune-related genes with pCR, most of the prediction performances were moderate and difficult to be implemented for clinical usage. The gene signatures and performances were also inconsistent across the studies since most of the studies were based on a few TNBC transcriptome datasets. Thus, it is still critical to have a deeper understanding of the mechanisms behind the responses of NAC of TNBC to develop more efficient experimental design and analysis strategies for TNBC biomarker discovery. 

Gene expression meta-analysis provides a valuable opportunity to integrate multiple datasets from different sources, increasing statistical power and accuracy of results. In this study, we conducted a large-scale gene expression meta-analysis of TNBC NAC responses based on 1739 samples from 16 datasets, the largest gene expression data collection for TNBC with NAC responses to our knowledge. We also performed another gene expression meta-analysis for relapse-free survival outcomes of TNBC using 7 out of 16 datasets. By integrating meta-analysis results from NAC and survival outcomes, we were able to dissect the TNBC cohort into four new NAC response subgroups and identify the genes and pathways associated with each subgroup. 

Our study provides a comprehensive overview of the gene expression changes associated with TNBC NAC responses and can help identify new treatment prediction biomarkers and therapeutic targets for TNBC patients. The results of this study could potentially lead to personalized treatment approaches for TNBC patients, improving their survival outcomes and overall quality of life.

## 2. Materials and Methods

### 2.1. Study Workflow

The study design and workflow are illustrated in Figure 1. We collected TNBC transcriptome data with NAC responses, including pathologic complete response (pCR) or residual disease (RD) outcomes, from 16 microarray/RNA-seq datasets. After preprocessing the microarray and RNA-seq data, we conducted differential testing for pCR outcome and association testing for survival outcome for each dataset. We then congregated the test results from each individual dataset using meta-analysis. The significant genes identified from pCR meta-analysis and survival meta-analysis were further divided into four subgroups. Finally, we performed functional enrichment analysis and gene network analysis for each subgroup. 

### 2.2. Gene Expression Datasets

A total of 16 publicly available neoadjuvant chemotherapy in breast cancer cohorts were selected for the study and listed in Table 1. The study analyzed a total of 1739 TNBC samples from the 16 studies. Seven cohorts contained pCR status and breast cancer disease-free survival information, while nine cohorts only contained pCR status. Since overall survival information was not available in most datasets, disease-free survival (DFS), recurrence-free survival, relapse-free survival, and distant disease-free survival were used as endpoints and treated equivalently in the survival analysis, with DFS used to represent them throughout the analysis. Among the 16 datasets, 5 were RNA-seq datasets generated by the Illumina platform, and 11 were microarray datasets generated by the Affymetrix platform. Except for the VUMC, all other datasets were downloaded from GEO using the GEOquery R package except for the VUMC dataset [32], which can be found in Appendix A. Table 1 provides detailed information on each dataset, including pCR and DFS information, references to the study, and the platform used.

### 2.3. Data Pre-Processing 

All microarray datasets were first normalized by the frozen robust multi-array analysis (fRMA) method [33], which performed background correction, normalization, and summarization, and removed the individual batch effect. This method was implemented in the fRMA R package. Samples that were not TNBC were filtered out, and for datasets lacking TNBC information, TNBCs were determined based on the status of estrogen receptor (ER), progesterone receptor (PR), and human epidermal growth factor receptor 2 (HER2). For datasets lacking ER, PR, and HER2 information, we used a two-component Gaussian mixture model with parameters estimated by R optim function to approximate the distribution of ER, PR, and HER2. The three Affymetrix probes 205225_at, 208305_at, and 216836_s_at were used to represent ER, PR, and HER2 gene expressions, respectively [34]. The posterior probability of negative expression status of ER, PR, and HER2 can be calculated by a mixture Gaussian model, respectively. We determined the negative status of ER, PR, and HER2 with their posterior probabilities larger than 0.5 [5,8]. Two microarray datasets were determined by the approximate distribution of ER, PR, and HER2, and other microarray datasets either included the status of ER, PR, and HER2, or provided TNBC information. The genes in the microarray datasets were represented by the probes with the largest interquartile range (IQR) statistics using findLargest function in genefilter R package. 

For the RNA-seq datasets, the counts and FPKM values were first converted to TPM values. Genes with a sum of TPM value of 0 were excluded from the datasets, and all genes in the datasets were then normalized by log2 transformation. 

### 2.4. Association Testing for Individual Dataset

To assess the association of gene expression with NAC response (pCR or RD), we performed logistic regression models with the probability of pCR as the outcome and genomic features as the main independent variable for the individual cohort. Age was adjusted in the model as the clinical variable. All logistics regression models were built using glm function in R. In the survival analysis, Cox proportional-hazard (PH) regression models implemented with coxph function in survival R package were used to test the association between gene expression and DFS outcome, including event and event-free in years. We also adjusted the clinical variable age in the Cox models. 

### 2.5. Meta-Analysis 

To integrate individual test results across different cohorts, we used the inverse-variance weighted random-effects model, which was implemented in the meta R package. The random-effects model was chosen because of the significant heterogeneities of cross-study datasets [35]. Association test results of pCR and survival analysis were used to perform meta-analysis separately. Specifically, in pCR meta-analysis, we used the coefficients and standard errors obtained from logistic regression models as estimates and standard errors of estimates as input to the meta-analysis model. The estimated effect sizes and standard errors from pCR meta-analysis were then re-scaled to compute odds ratios for every one-unit increase in expression values. In the survival meta-analysis, we used the log hazard ratios and standard errors from Cox PH regression models as input. Similarly, the hazard ratios were computed by the estimated effect sizes and standard errors from the survival meta-analysis. We included genes present in least three cohorts that appeared in the pCR and survival meta-analysis, respectively. 

To select the significant genes associated with pCR and DFS outcome, we first divided the genes into four quadrants based on the results from pCR meta-analysis and survival meta-analysis illustrated in Figure 1. In gene list quadrant one, we selected genes with positive effect sizes in pCR meta-analysis and with negative effect sizes in survival meta-analysis. This quadrant represents a favorable DFS outcome with pCR status. Similarly, quadrants two, three, and four represent the genes associated with favorable DFS outcomes with RD status, unfavorable DFS outcomes with RD status, and unfavorable DFS outcomes with pCR status, respectively. 

### 2.6. Pathway and Functional Enrichment Analysis

Gene set enrichment analysis (GSEA) implemented in Bioconductor/R package fgsea was applied for the pCR and survival meta-analysis gene lists [36]. The functional enrichment analyses were performed for gene lists in each of the four quadrants, respectively, using over-representation analysis [37], which was implemented in clusterProfiler 4.0 Bioconductor/R package [38]. Briefly, this method determined whether biological processes that were over-represented in the gene list of interest using *p*-values calculated by hypergeometric distribution and adjusted by the Benjamini–Hochberg (BH) method to calculate false discovery rate (FDR). The pathway collection for both GSEA and functional enrichment analysis is BioCarta subset, KEGG subset, PID subset, Reactome subset, and WikiPathways subset of Canonical pathways (C2:CP) from the Molecular Signatures Database (MSigDB) [39], which can be accessed by the msigdbr Bioconductor/R package. The minimum and maximum sizes of gene sets used for analysis were set to 10 and 500, respectively. FDR 0.25 was used for the significant threshold for both GSEA and functional enrichment analysis. 

### 2.7. Protein–Protein Interaction (PPI) Network Analysis

To assess the interactive associations among the four-quadrant gene lists at the protein level, we performed PPI network analysis using NetworkAnalyst webtool (version 3.0) [40], which can be accessed at http://www.networkanalyst.ca (accessed on 2 February 2023). NetworkAnalyst provides degree and betweenness to help identify the highly interactive hub nodes. The degree is used to measure the number of connections of a node to other nodes, and the betweenness measures the number of shortest paths to a node. We uploaded gene lists to the webtool and performed the PPI network analysis with International Molecular Exchange (IMEx) Interactome database [41]. The networks were built either based on the original seed proteins through the first-order network or the minimum network by trimming the first-order network to keep the essential nodes that connected the seed nodes. 

## 3. Results

### 3.1. Meta-Analysis Results of NAC Response

We performed a meta-analysis based on association testing results from 16 datasets to assess the association of gene expression with TNBC NAC response (pCR vs. RD). Table 2 lists the top 10 positively associated genes with pCR (upregulated in pCR group) and the top 10 negatively associated genes with pCR (upregulated in RD group), while the complete gene list of pCR meta-analysis can be found in Appendix A. We observed a significantly unbalanced association pattern, with 1656 genes positively associated with pCR and only 70 genes negatively associated with pCR out of 1726 genes passing the FDR threshold of 0.05. Figure 2 displays the top 10 pathways with positive normalized enrichment score (NES) (positively associated with pCR) and the top 10 pathways with negative NES. Our pathway analysis suggests that cell cycle/mitotic and immune-related pathways are associated with pCR, while metabolism-related pathways are associated with RD. The complete GSEA analysis results can be found in Appendix A.

### 3.2. Meta-Analysis Results of DFS 

We also performed a meta-analysis based on association testing results with TNBC DFS survival outcomes from seven datasets. Table 3 lists the top 10 positively associated genes with survival (upregulated gene associated with higher survival risk) and the top 10 negatively associated genes with survival (upregulated gene associated with lower survival risk). Due to the limited availability of TNBC datasets with survival outcomes, we used nominal *p*-values instead of FDR to detect significant genes. The complete gene list of survival meta-analysis and pathway list from GSEA analysis can be found in Appendix A. The unbalanced association pattern observed in the pCR meta-analysis was also evident in the survival meta-analysis, with 888 genes having a hazard ratio less than 1 (upregulated gene expression associated with lower survival risk) and only 91 genes having a hazard ratio greater than 1 out of 979 genes passing the nominal *p*-value 0.05. Figure 3 displays the top 10 pathways with positive NES (associated with higher survival risk) and the top 10 pathways with negative NES. Our pathway analysis suggests that RNA splicing and immune-related pathways are associated with favorable survival outcomes, while cell junction and metabolism-related pathways are associated with unfavorable survival outcomes. 

### 3.3. Integration of pCR and DFS Meta-Analysis Results

The pCR meta-analysis and DFS meta-analysis show substantial overlap in cell cycle and immune pathways; these findings are expected as pCR was shown to be a surrogate for survival outcomes for TNBC (Appendix A) [21,42]. However, the discrepancy between the two analysis results provides the opportunity to discover new biomarkers and therapeutic targets for TNBC treatments. We further dissected the significant genes from pCR and DFS meta-analysis results into four quadrants based on their association with pCR and DFS. The criteria to select genes for each quadrant were either FDR < 0.05 in pCR meta-analysis or *p*-value < 0.05 in DFS meta-analysis. Since there could potentially be a large number of genes selected in quadrant one, we used a more stringent criterion requiring both FDR < 0.05 in pCR meta-analysis and *p*-value < 0.05 in survival meta-analysis to select genes for quadrant I. The complete gene lists for each quadrant can be found in Appendix A. We discuss the downstream analyses, including functional enrichment analysis and PPI network analysis, in the following subsections.

### 3.4. Quadrant I: Genes Associated with pCR and Lower DFS Risk

The number of significant genes in quadrant I is dominant compared to the other three quadrants (Figure 4). These genes are associated with favorable clinical outcomes of TNBC, including pCR and lower DFS risk. The functional enrichment analysis results (Appendix A and Figure 5A) show that immune and mRNA preprocessing-related pathways are enriched in the top-ranking list, in addition to numerous cell cycle and DNA repair pathways which is concordant with the GSEA analysis results for pCR and DFS meta-analysis. Figure 5B displays the PPI network using the Quadrant I significant genes as the seeds. *SUMO2*, *CUL3*, *CAND1*, *GRB2,* and *HDAC4* are the top hub genes with the most interactions with other genes. 

### 3.5. Quadrant II: Genes Associated with RD and Lower DFS Risk

The genes in quadrant II are relatively sparse and associated with favorable long-term clinical outcomes of TNBC, including lower DFS risk, despite their association with unfavorable short-term clinical outcomes (RD). These tumors likely represent slower growing chemo-resistant tumors that have favorable prognosis. Only a few pathways passed the FDR 0.25 threshold and are related to the extracellular matrix and elastic fiber (Appendix A and Figure 6A). *CCND1*, *ZBTB16*, *FLNC*, and *PPP2R2B* are displayed as major hub genes in PPI network (Figure 6B).

### 3.6. Quadrant III: Genes Associated with RD and Higher DFS Risk

Quadrant III contains the second-largest number of genes among the four quadrants and is associated with unfavorable clinical outcomes, including RD and higher DFS risk. Functional enrichment analysis shows that the G-protein-coupled receptor (GPCR) pathways are the major significant functional gene sets (Appendix A and Figure 7A). The hub genes include *H4C9*, *H3C12*, *CFTR*, *SNCA*, and *PRKN* (Figure 7B).

### 3.7. Quadrant IV: Genes Associated with pCR and Higher DFS Risk

Quadrant IV has the least number of genes among the four quadrants, which are associated with pCR and high survival risk. 3q29 copy number variation syndrome gene set is significantly enriched with the quadrant IV genes (Appendix A and Figure 8A). *RAD21*, *SRPK1*, *H3C3,* and *ISG15* are among the most connected genes within the quadrant IV PPI network (Figure 8B). 

## 4. Discussion

TNBC is still being treated as a single disease, and NAC is the common treatment strategy for TNBC, which involves administering chemotherapy before surgery to shrink the tumor and, potentially, improve surgical outcomes. However, the heterogeneous responses of TNBC patients for NAC indicate the need for precision medicine, including the identification of biomarkers to predict NAC response and inform personalized treatment strategies. 

The molecular mechanisms underlying NAC response in TNBC are complex and involve interactions between the tumor and the host immune system, as well as between the tumor and the chemotherapy drugs. In the past, numerous studies have been conducted to discover the NAC response mechanisms and biomarkers, but the inconsistent results and inconclusive prediction performances show that it is still an open research area [23,24,25,26,27,28,29,30,31]. The large-scale gene expression meta-analyses we conducted not only increase the statistical power to detect more signals associated with NAC response and survival outcome, also provide an innovative way to integrate both NAC response and survival outcome to gain more biological insight into potential NAC response mechanisms. 

The immune and proliferation pathways are significantly enriched at the top pathway list in pCR meta-analysis, and immune pathways are also strongly associated with lower DFS risk in the survival meta-analysis. Those results verified the outcomes from previous studies about the association of immune and proliferation signatures with pCR and survival outcomes of TNBC patients [23,30,43,44,45]. RNA splicing-related pathways are also among the top pathways associated with both pCR and lower DFS risk, which was shown the important role of controlling the proliferation of TNBC cells [46]. 

*SLAMF7* and *CCND1* are the most significantly associated genes with pCR (HR 1.419, adjusted *p*-value 2.95 × 10^−10^) and RD (HR, 0.770, adjusted *p*-value 7.85 × 10^−6^), respectively, in the meta-analysis. *SLAMF7* encodes surface antigen CD319, which is a stable marker of both normal and malignant plasma cells in multiple myeloma [47]. Besides its expression on myeloma cells, *SLAMF7* is also present in immune cells, particularly in natural killer (NK) cells, where it serves as a costimulatory receptor to stimulate innate and adaptive immunity. *SLAMF7* was included in the immune gene signatures to predict TNBC NAC response [29,48], and it was also used as a target by a nanoconjugate system to activate an immune response for breast cancer [49]. *CCND1* is associated with RD and plays an important role in cell cycle regulation. It encodes a protein that forms a complex with cyclin-dependent kinase 4 (CDK4) or CDK6, promoting the progression of cells through the G1 phase of the cell cycle. Aberrant expression of *CCND1* has been found in a variety of human cancers, including breast cancer [50,51,52]. In breast cancer, *CCND1* is frequently overexpressed due to gene amplification or other mechanisms, and its overexpression has been associated with the development and progression of the disease [53,54,55,56]. *CCND1* is associated with luminal breast cancers and amplifications have been found in chemoresistant post-treatment ER+ tumors [57].

A prominent result from NAC response and DFS meta-analysis is the extremely unbalanced significant upregulated genes associated with pCR and lower survival risk, which could be caused by the sheer number of differentially expressed transcripts associated with immune cells and large number or transcript involved in cell cycle progression. This phenomenon partially explains the challenge of achieving the satisfying specificity for TNBC NAC signatures [26,27,29] since the number and effect size of the positively associated genes overwhelms the negatively associated genes. 

To maximally utilize the information from NAC response and DFS meta-analysis, we further divided the gene association results into four quadrants. The explanation of quadrant I and III is straightforward. Genes in quadrant I are associated with pCR and lower survival risk, and the prevalence of immune and cell cycle genes associated with chemotherapy. Genes in quadrant III are associated with RD and higher survival risk, and G-protein-coupled receptors (GPCRs) pathways stand out in enrichment analysis, which is not at the top-ranking list in NAC response or survival analysis alone. 

Quadrant II genes are associated with RD and lower survival risk. Those genes could be linked to a group of TNBC patients who are insensitive to NAC but with less aggressive growth patterns of tumors. Luminal androgen receptor (LAR) subtype TNBC patients defined by our TNBC type are less responsive to NAC compared to other TNBC subtypes [58,59], but breast cancer patients with positive androgen receptor (AR) expression have favorable long-term clinical outcomes, including overall survival and recurrence [60,61]. In our recent multi-omics analysis of TNBC, we showed that LAR subtype has a genetic dependency on *CCND1* [9], which happened to be the hub gene within quadrant II. 

It seems to be challenging to explain the genes in quadrant IV. However, a small group of patients with micrometastases in sentinel lymph modes could have a higher risk of recurrence even with the pCR outcome in NAC [62,63]. 3q29 copy number variation syndrome pathway is far more significantly enriched than other pathways in quadrant IV genes. Previously, 3q29 region was shown as an amplified region in basal-like breast cancer with BRCA1 mutation [64], and it was also identified as a significantly altered region in the metastatic breast cancer cohort [65]. 

With this large-scale gene expression meta-analysis, several conclusions are summarized as the following. First, different and appropriate clinical endpoints need to be considered for NAC response. Survival outcomes were shown to be a more effective endpoint to evaluate the combination of durvalumab, which is an immune-checkpoint inhibitor, with NAC for early-stage TNBC in phase II clinical trial [66]. As we demonstrated in the four-quadrants analysis of NAC response and survival outcomes, integration of both short-term and long-term clinical outcomes is more meaningful in guiding clinical decisions. Residual cancer burden (RCB), which is a continuous index to measure treatment responses based on primary tumor and nodal metastases, is considered the better endpoint to replace pCR for the neoadjuvant treatment responses in clinical trials [67,68]. 

Second, the multi-modal prediction models need to be implemented for NAC prediction. The most available prediction models were built based on gene expression data. Recently, the prediction model using multi-modal data, such as DNA mutation, gene expression, copy number, digital pathology, and imaging, etc., achieved better prediction performance for multiple cancer prognosis and treatment prediction, including breast cancer NAC [69,70,71,72]. The research community needs to make an effort to collect multi-modal data for TNBC clinical trials. The unbalanced signals issue from gene expression alone could be solved using multi-modal data of TNBCs. 

Third, racial disparities need to be included during the biomarker discovery and NAC prediction model building for TNBC. The incidence of TNBC compared to other types of breast cancer has been estimated to be approximately 28% among African American women and 20% among Hispanic women compared to 12% for non-Hispanic white patients [73,74]. Compared to Whites, African Americans and Hispanics were more likely to present with regional or metastatic disease and have comorbidity among breast cancer patients. Based on the National Cancer Data Base, among TNBC patients who received NAC, black women had significantly lower rates of pCR compared with white women [75]. Current evidence from multiple studies suggests that both socioeconomic diversity and tumor biology contribute to the racial disparities in TNBC [76]. There is an urgent need to include socioeconomic status and socioeconomic-genomic interaction effects in prediction models. 

## 5. Conclusions

In conclusion, TNBC is a heterogeneous disease that requires precision medicine approaches, including the identification of biomarkers to predict NAC response and inform personalized treatment strategies. The molecular mechanisms underlying NAC response in TNBC are complex, and large-scale gene expression meta-analyses provide a way to integrate both NAC response and survival outcome to gain more biological insight. The results of this study highlight the importance of considering appropriate clinical endpoints and multi-modal prediction models for NAC prediction, as well as racial disparities during biomarker discovery and NAC prediction model building for TNBC. Overall, this study contributes to the ongoing efforts to improve the management of TNBC and personalize treatment strategies for patients with this disease.

## Figures and Tables

**Figure 1 cancers-15-02194-f001:**
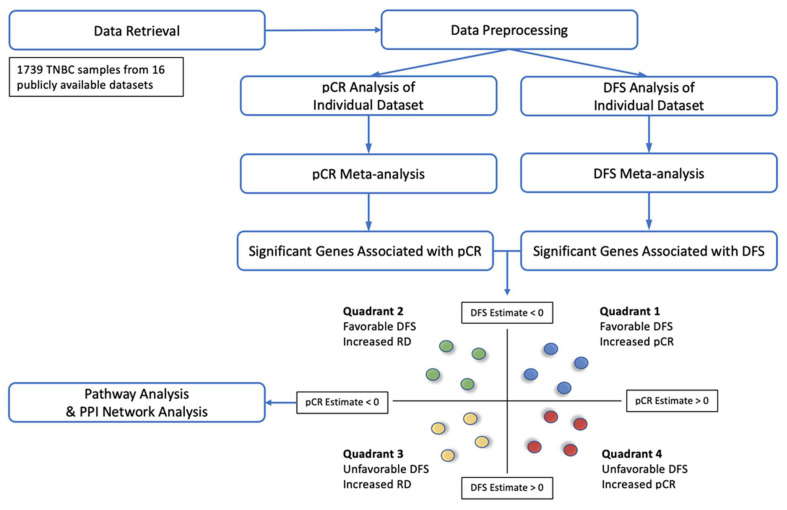
The study design and workflow of the transcriptome meta-analysis of TNBC.

**Figure 2 cancers-15-02194-f002:**
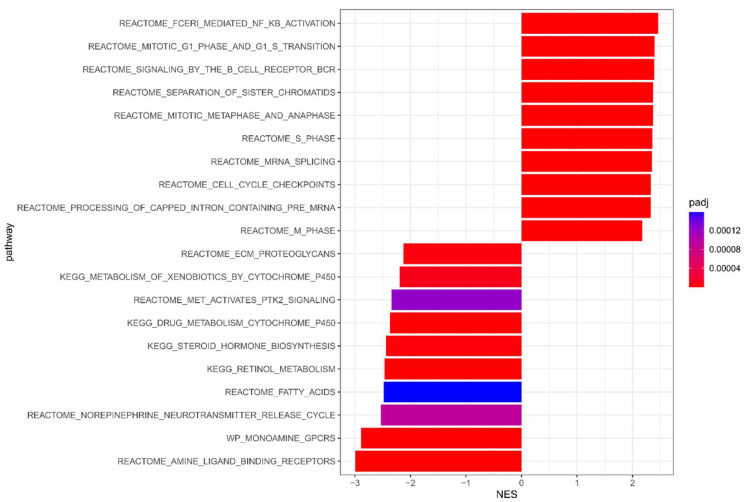
Top pathways of GSEA analysis for TNBC NAC response meta-analysis.

**Figure 3 cancers-15-02194-f003:**
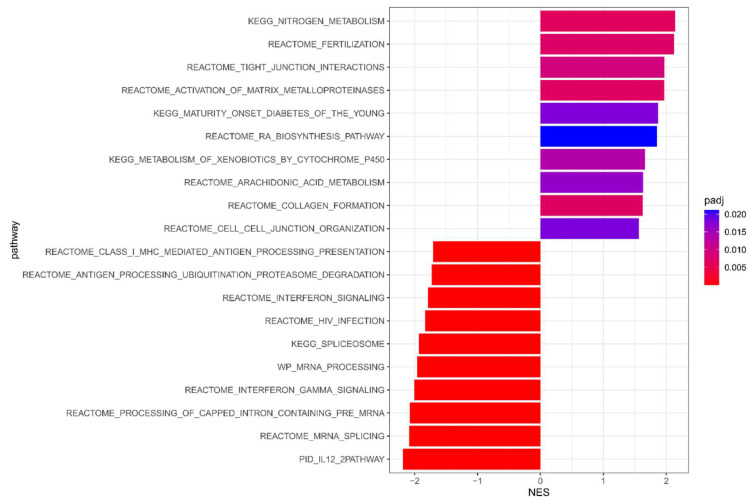
Top pathways of GSEA analysis for TNBC DFS meta-analysis.

**Figure 4 cancers-15-02194-f004:**
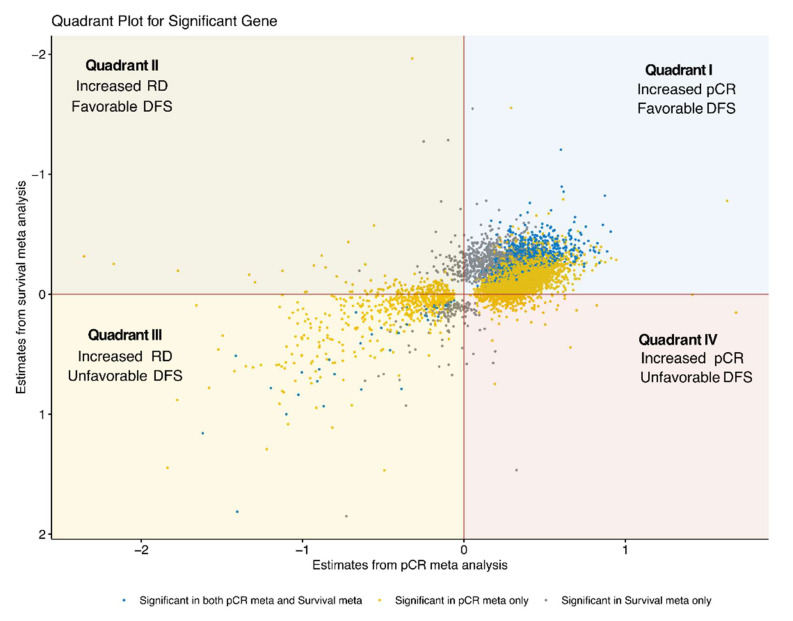
Quadrant plot of significant genes from integration of TNBC NAC and DFS meta-analysis.

**Figure 5 cancers-15-02194-f005:**
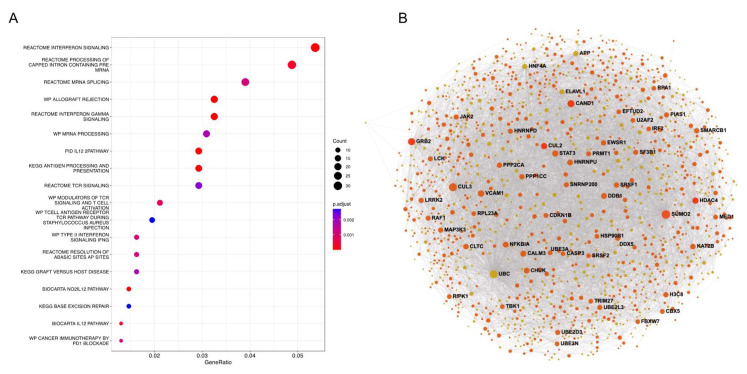
(**A**) Top pathways from functional enrichment analysis of quadrant I genes (**B**) PPI network of quadrant I genes.

**Figure 6 cancers-15-02194-f006:**
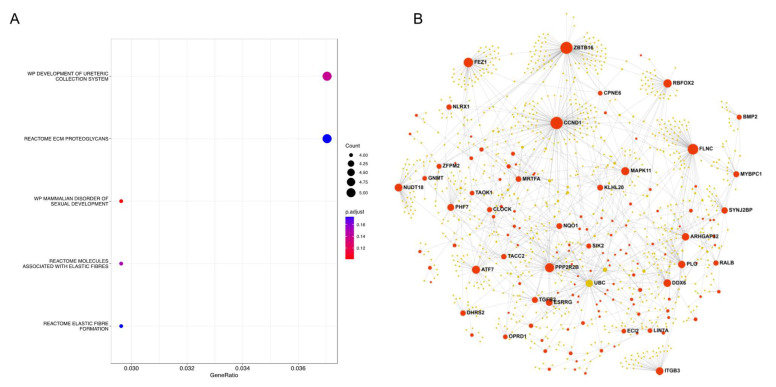
(**A**) Top pathways from functional enrichment analysis of quadrant II genes. (**B**) PPI network of quadrant II genes.

**Figure 7 cancers-15-02194-f007:**
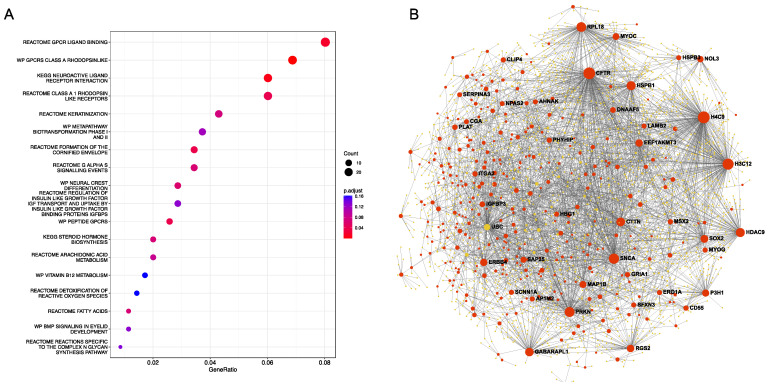
(**A**) Top pathways from functional enrichment analysis of quadrant III genes. (**B**) PPI network of quadrant III genes.

**Figure 8 cancers-15-02194-f008:**
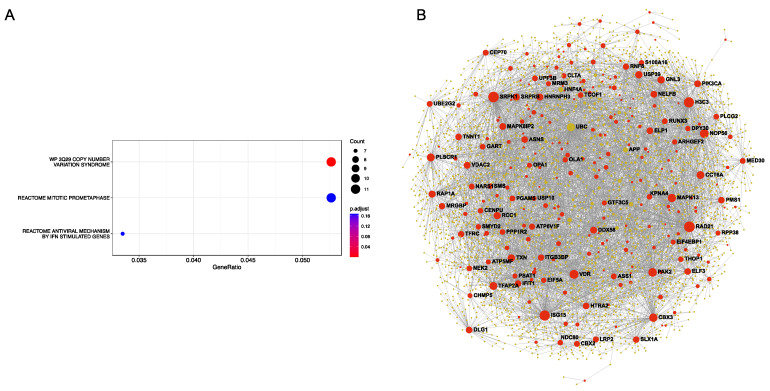
(**A**) Top pathways from functional enrichment analysis of quadrant IV genes. (**B**) PPI network of quadrant IV genes.

**Table 1 cancers-15-02194-t001:** Patient and characteristics of gene expression datasets.

Datasets	Type	Numbers of Sample		pCR			DSF Event		Median Follow-Up (sd, Range), Years	Mean Age (sd, Range)
		Total	TNBC	Yes	No	NA	Yes	No		
GSE16446	Microarray	120	86	10	73	3	18	68	2.9 (1.5, 0.2–5.9)	/
GSE20271	Microarray	178	59	13	46	/	/	/	/	51 (10.8, 29–74)
GSE22093	Microarray	103	39	16	23	/	/	/	/	46.7 (10.2, 31–67)
GSE18864	Microarray	84	24	8	16	/	/	/	/	49.8 (10.3, 29–68)
GSE20194	Microarray	278	71	25	46	/	/	/	/	50.4 (10.8, 29–75)
GSE25055	Microarray	310	119	40	78	1	37	82	2.1 (1.7, 0.1–7.4)	49.6 (10.8, 28–75)
GSE25065	Microarray	198	63	22	36	5	22	41	2.8 (1.5, 0.4–6.1)	49.1 (11.2, 24–72)
GSE18728	Microarray	61	22	6	16	/	/	/	/	/
GSE41998	Microarray	279	145	47	83	15	/	/	/	/
GSE32646	Microarray	115	26	10	16	/	/	/	/	54.4 (12.5, 28–70)
GSE164458	RNA-Seq	482	482	236	246	/	/	/	/	/
VUMC	RNA-Seq	145	45	29	16	/	8	37	2.6 (0.8, 0.7–5.8)	50.6 (10.8, 29–75)
GSE154524	RNA-Seq	389	389	210	179	/	115	274	5.3 (2, 0.1–8.1)	/
GSE22226	RNA-Seq	150	27	7	20	/	11	16	3.6 (2.2, 0.5–6.7)	47 (8, 33.5–63.2)
GSE192341	RNA-Seq	87	52	21	29	2	10	42	1.8 (1.2, 0.4–4.2)	44.6 (11.7, 26–73)
GSE163882	RNA-Seq	222	90	38	52	/	/	/	/	54.1 (11.5, 23–74)

**Table 2 cancers-15-02194-t002:** Top 10 positive and negative associated genes with pCR outcomes.

Gene	Odds Ratio	Estimate	se	z-Value	*p*-Value	FDR	Direction *	Number of Datasets
* Positive associated genes *								
SLAMF7	1.419	0.350	0.045	7.792	6.62 × 10^−15^	2.95 × 10^−10^	+++++-++++-+-+++	16
GBP5	1.398	0.335	0.043	7.736	1.02 × 10^−14^	2.95 × 10^−10^	+++++-+-++	10
GZMB	1.356	0.305	0.041	7.461	8.59 × 10^−14^	1.65 × 10^−9^	++++-+++++++-+++	16
CD274	1.590	0.464	0.063	7.404	1.32 × 10^−13^	1.91 × 10^−9^	+++++++-++	10
GBP4	1.474	0.388	0.054	7.189	6.54 × 10^−13^	7.56 × 10^−9^	+++++++-++	10
NUP210	1.599	0.470	0.067	6.972	3.14 × 10^−12^	3.02 × 10^−8^	++++--++++++++++	16
NOL7	2.017	0.702	0.101	6.947	3.73 × 10^−12^	3.08 × 10^−8^	++++-+++++++++++	16
MCM5	1.749	0.559	0.082	6.843	7.76 × 10^−12^	5.61 × 10^−8^	+++-+++++++++++	15
NKG7	1.348	0.298	0.044	6.809	9.84 × 10^−12^	6.32 × 10^−8^	++++++++++--+++	15
CXCL10	1.244	0.218	0.032	6.775	1.25 × 10^−11^	7.20 × 10^−8^	+++++-+++++-+++	15
* Negative associated genes *								
CCND1	0.770	−0.261	0.046	−5.695	1.24 × 10^−8^	7.85 × 10^−6^	-+-+-+----------	16
LINC00622	0.635	−0.454	0.086	−5.260	1.44 × 10^−7^	4.49 × 10^−5^	---+---+	8
CTSF	0.683	−0.381	0.074	−5.124	2.98 × 10^−7^	8.13 × 10^−5^	----+---++-++--+	16
SLC4A11	0.814	−0.206	0.044	−4.638	3.51 × 10^−6^	5.23 × 10^−4^	----+--+--	10
ZNF697	0.651	−0.429	0.095	−4.535	5.77 × 10^−6^	7.57 × 10^−4^	--------++	10
AHNAK2	0.825	−0.193	0.043	−4.500	6.80 × 10^−6^	8.68 × 10^−4^	-+--+--+-------+	16
OPLAH	0.800	−0.223	0.052	−4.260	2.05 × 10^−5^	2.04 × 10^−3^	----++---------+	16
MAPRE3	0.702	−0.353	0.086	−4.089	4.34 × 10^−5^	3.69 × 10^−3^	-+--+----------+	16
MGAM	0.767	−0.266	0.066	−4.047	5.19 × 10^−5^	4.26 × 10^−3^	--------------+	15
MSX1	0.755	−0.282	0.071	−3.971	7.15 × 10^−5^	5.29 × 10^−3^	---+-----------+	16

* + repressents postive association in the single dataset; - represents negtative association in the single dataset.

**Table 3 cancers-15-02194-t003:** Top 10 positive and negative associated genes with DFS outcomes.

Gene	Hazard Ratio	Estimate	se	z-Value	*p*-Value	FDR	Direction *	Number of Datasets
* Positive associated genes *								
MST1L	1.367	0.312	0.078	3.985	6.74 × 10^−5^	8.70 × 10^−2^	+-++	4
LINC00905	2.043	0.715	0.212	3.369	7.56 × 10^−4^	2.55 × 10^−1^	+++	3
MYBPH	1.165	0.153	0.048	3.163	1.56 × 10^−3^	3.13 × 10^−1^	+-+-+++	7
FAM20C	1.372	0.316	0.100	3.151	1.63 × 10^−3^	3.16 × 10^−1^	+++-+	5
CLDN16	1.145	0.135	0.043	3.120	1.81 × 10^−3^	3.35 × 10^−1^	+-++++	6
SCGB2A1	1.098	0.093	0.030	3.090	2.00 × 10^−3^	3.46 × 10^−1^	+++++++	7
TGM5	1.132	0.124	0.041	3.045	2.33 × 10^−3^	3.46 × 10^−1^	++++++	6
VPS50	2.202	0.790	0.265	2.985	2.84 × 10^−3^	3.46 × 10^−1^	+++-	4
LINC00032	1.302	0.264	0.090	2.933	3.35 × 10^−3^	3.69 × 10^−1^	+-+	3
TBC1D21	4.330	1.466	0.505	2.902	3.70 × 10^−3^	3.77 × 10^−1^	+-+	3
* Negative associated genes *								
PPP1R12A	0.628	−0.465	0.107	−4.331	1.48 × 10^−5^	8.70 × 10^−2^	-------	7
AKAP5	0.627	−0.467	0.110	−4.236	2.28 × 10^−5^	8.70 × 10^−2^	---+--	6
SLAMF7	0.818	−0.200	0.049	−4.077	4.57 × 10^−5^	8.70 × 10^−2^	-------	7
VEZF1	0.640	−0.446	0.110	−4.075	4.61 × 10^−5^	8.70 × 10^−2^	-------	7
OGG1	0.586	−0.534	0.132	−4.060	4.90 × 10^−5^	8.70 × 10^−2^	-----+-	7
LRRK2	0.651	−0.429	0.106	−4.048	5.16 × 10^−5^	8.70 × 10^−2^	-----	5
RBMXL1	0.596	−0.517	0.131	−3.943	8.03 × 10^−5^	8.70 × 10^−2^	---+-	5
HERPUD1	0.670	−0.401	0.104	−3.837	1.25 × 10^−4^	9.95 × 10^−2^	------	6
PDCD1LG2	0.729	−0.316	0.083	−3.795	1.48 × 10^−4^	1.12 × 10^−1^	---+---	7
TXNDC5	0.639	−0.448	0.118	−3.792	1.49 × 10^−4^	1.12 × 10^−1^	---	3

* + repressents postive association in the single dataset; - represents negtative association in the single dataset.

## Data Availability

RNAseq normalized data matrix for the VUMC cohort provided in Appendix A.

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
