# Peer review of "Transcriptome Meta-Analysis of Triple-Negative Breast Cancer Response to Neoadjuvant Chemotherapy"

_cancers, 2023, doi:10.3390/cancers15082194_

Round 1
Reviewer 1 Report
This study was conducted by a team of researchers from various institutions in the USA. The team have the complementary expertise in the cancer biology and genomics. They analyzed the gene expression profiles of triple-negative breast cancer patients who underwent neoadjuvant chemotherapy to identify potential biomarkers for predicting treatment response. This is a well organized manuscript. A few general comments:
1. It will be better to have more discussion about why is it important to study its response to neoadjuvant chemotherapy? and What are the potential clinical implications of this study, and how could it impact the treatment of triple-negative breast cancer patients in the future.
2. Since the author include more microarray and RNAseq data in this meta-analysis. How did the researchers to avoid the platform bias from different technologies?
3. Authors performed network analysis It would be preferable to make the gene's name more prominent. In addition, I believe that Figures 5 through 8 could be merged into a single figure since they contain similar information.
Author Response
We appreciate the reviewer’s time and thoughtful critique and have provided a point-by-point rebuttal addressing each of the reviewer's concerns below.
(1)We appreciated the reviewer's comment and have modified the introduction to, “Neoadjuvant chemotherapy is the standard for newly diagnosed TNBC. While ~30% of patients respond and have favorable outcomes, there are a large portion of patients that must endure the toxicities associated with chemotherapy. The ability to identify those patients unlikely to respond could give clinicians guidance to enroll patients on alternative compounds in clinical trials.”
(2)To avoid platform bias in microarray datasets, we only included Affymetrix datasets that were based on the U133A platform and contained Affymetrix probes 205225_at, 208305_at, and 216836_s_at were used to represent ER, PR, and HER2 gene expressions. Furthermore, each analysis is separate and independent from each other datasets with the results integrated for the meta-analysis.
(3)We agree with the reviewer and have increased font size and improved resolution for Figures 5-8. However, merging Figures 5-8 would be too difficult for the reader to interpret and therefore we have kept them as separate figures.
Reviewer 2 Report
Zhang and colleagues report in this study a large-scale gene expression meta-analysis of neoadjuvant chemotherapy (NAC) responses and disease-free survival (DFS) outcomes of triple-negative breast cancer (TNBC) using 16 public datasets. Based on this meta-analysis they can stratify TNBC patients into new subgroups and identify subgroup-specific gene pathway enrichment that holds potential for prognostic application. However, the molecular subtyping in TNBC or in pan-cancers based on publicly available transcriptomic datasets (eg: TCGA), has been recently and extensively studied. This type of study generates vast amounts of data suggesting a transcriptional association with clinical outcomes, which could be an excellent starting point for a study. It will be interesting and of particular importance to verify such predictions (aberrant genes/pathways) using local cohort or to test by functional experimental assays. As the authors identify interesting associations between cell cycle pathways, RNA splicing and favorable clinical outcomes, how these pathways impact specifically TNBC treatment is largely unknown and untested.
Specific concerns:
1. From the 16 datasets, the exact numbers of patients with TNBC (post-filtering) should also be included in Table 1.
2. For datasets that lack pathological examination, the authors compute the TNBC status based on the expression of ER, PR, and HER2, or a model that approximates the distribution of the expression of these receptors. Could you please include the references? And could you provide evidence that these estimates closely mimic the actual clinical pathological examination? For example, comparing pathological status with computational estimates using datasets with both information.
3. Do all 11 microarrays use the same probes for ER, PR, and HER2?
4. How to explain the pathways upregulated in pCR (pathological complete response) as cell cycle/mitotic pathways but not apoptosis/cell death-related ones?
5. While the authors say “The pCR meta-analysis and DFS meta-analysis show a similar trend as expected”, there is no overlap in top GESA pathways between the high pCR group and low survival risk group.
Author Response
We appreciate the reviewer’s time and thoughtful critique and have provided a point-by-point rebuttal addressing each of the reviewer's concerns below.
(1)To address this problem, we included both total samples (pre-filtering) and TNBC samples (post-filtering) in Table 1.
(2)We have previously published and extensive analysis across 11 datasets (PMID: 21633166, Table 1) using bimodal filtering on expression and comparing to clinical pathological calls. While the correspondence varied between datasets, the overall false-positive rates were 1.7%, 1.7%, and 0.9% for ER, PR, and HER2, respectively, demonstrating that bimodal filtering of data sets by mRNA expression is a reliable method for identifying TNBC tumors from data sets lacking IHC information.
Among all 11 microarrays, only two datasets were determined by the approximate distribution of ER, PR, and HER2. Other datasets either included the status of ER, PR, and HER2, or provided TNBC information.
(3)The three Affymetrix probes 205225_at, 208305_at, and 216836_s_at were used to represent ER, PR, and HER2 gene expressions, respectively.
(4)The pathways upregulated in pCR reflect highly proliferative tumors with increased dependency on cell cycle pathways that are targeted by DNA-damaging and microtubule targeting agents in standard of care (Adriamycin, Cytoxan and Taxanes). These pathways exist in the pre-treatment biopsies and the increase of apoptosis/cell-death is expected to occur after treatment. We anticipate those patients responding to chemotherapy would have increased cell death pathways in an on-treatment biopsy. However, those biopsies are typically not performed and for responding patients (pCR), there will be no evidence of tumor cells in the surgical specimen.
(5)We apologize for this confusion, as the figures only show the top 10 significant pathways from each analysis. here were so many pathways that were significant for the pCR analysis, and the overlap can be seen in the supplemental tables (Table S2 and Table S3). We have modified the sentence to, “The pCR meta-analysis and DFS meta-analysis show substantial overlap in cell cycle and immune pathways, these findings are expected as pCR was shown to be a surrogate for survival outcomes for TNBC (Table S2 and S3).”
Round 2
Reviewer 2 Report
Thanks to the authors for their efforts to address most of the concerns. It will be great to see how the predictions by Meta-analysis in this report be verified in wet experiments in future studies.